# VLAAR: Vision-Language Attribute-Aware Router for Pedestrian Attribute Recognition

## Abstract

Pedestrian attribute recognition aims to identify multiple semantic attributes of individuals from visual data, a task critical for surveillance applications. However, existing methods often overlook the heterogeneity of pedestrian attributes and lack mechanisms for effectively modeling inter-attribute relationships. This paper proposes VLAAR, a parameter-efficient fine-tuning method for pedestrian attribute recognition that leverages the mixture-of-experts framework. Building upon a pre-trained CLIP model, our approach employs lightweight expert modules, forming a pool of specialized networks. At its core, our dual-input routing mechanism concurrently processes visual features alongside semantic cues derived from natural language prompts, guiding expert selection effectively. This dynamic routing facilitates the optimal allocation and efficient processing of complex attribute information while preserving computational efficiency. Extensive evaluations on image and video benchmarks demonstrate state-of-the-art performance for multi-label attribute recognition in surveillance and re-identification systems.

## 1 Introduction

Pedestrian attribute recognition (PAR) is a fundamental task in computer vision that involves the simultaneous identification of multiple semantic attributes of individuals from visual data. These attributes span appearance characteristics (e.g., clothing style, color), accessories (e.g., backpack, glasses), demographic information (e.g., age, gender), and behavioral patterns (e.g., walking, running). As a key component in intelligent surveillance, person re-identification, and human–computer interaction, PAR has attracted considerable attention from both academia and industry in recent years. While deep learning has significantly advanced PAR beyond early hand-crafted feature methods (Zhang et al., 2014), key challenges remain. First, the inherently multi-label nature of PAR requires models to capture attributes with highly diverse visual salience. Second, attributes often exhibit complex interdependencies that are difficult to model using conventional architectures. Third, the high computational cost of fully fine-tuning large-scale models poses limitations for deployment in resource-constrained settings.

Parameter-efficient fine-tuning (PEFT) has emerged as promising solutions to address computational constraints by adapting pre-trained models with minimal additional parameters. Methods such as adapters (Houlsby et al., 2019), prompt tuning (Lester et al., 2021), and LoRA (Hu et al., 2022) have demonstrated impressive results across various vision tasks. However, these approaches typically employ uniform adaptation strategies that fail to adequately address the inherently heterogeneous nature of pedestrian attributes. Moreover, they lack mechanisms to explicitly capture inter-attribute relationships, which are crucial for accurate recognition. Mixture-of-experts (MoE) architectures (Riquelme et al., 2021; Fedus et al., 2022) offer a compelling alternative by conditionally activating specialized sub-networks for different inputs. Various implementations (Lepikhin et al., 2021; Riquelme et al., 2021) with different routing mechanisms (Shazeer et al., 2017; Zhou et al., 2022) have shown success in large language models and vision transformers. Nevertheless, existing MoE methods typically allocate experts in a purely data-driven manner, relying only on input features. This overlooks the rich semantic structure among pedestrian attributes, which could be exploited to guide more effective specialization and routing.

To address these challenges, we propose VLAAR (Vision-Language Attribute-Aware Router), a parameter-efficient mixture-of-experts framework featuring architecturally diverse expert specialization and a dual-modality routing mechanism. VLAAR integrates three distinct expert types: (i) standard feedforward networks for basic transformations, (ii) advanced multi-layer networks with residual connections for complex hierarchical patterns, and (iii) convolutional experts for local spatial relationships. Unlike conventional approaches, VLAAR strategically allocates computational resources based on both visual content and attribute semantics. Specifically, pedestrian attributes are organized into semantically meaningful groups using language prompts, enabling effective cross-modal knowledge transfer from CLIP and improving both image- and video-based attribute recognition. Extensive experiments show that VLAAR consistently outperforms existing parameter-efficient fine-tuning (PEFT) methods on standard benchmarks, while maintaining minimal parameter overhead. Our contributions are summarized as follows:

- We propose VLAAR, a novel parameter-efficient fine-tuning method which dynamically selects expert modules by effectively integrating visual features and language-derived semantic cues. This enables customized processing tailored specifically for diverse attribute types.

- We design an Attribute-Aware Mixture of Expert module, characterized by attribute-specific expert specialization and dual-modality routing. By selectively activating only the Top-$k$ relevant experts per input, this module both enhances inter-attribute reasoning.

- Extensive experiments on diverse dataset demonstrate that VLAAR consistently outperforms existing parameter-efficient fine-tuning methods, achieving superior results.

## 2 RELATED WORK

### 2.1 PEDESTRIAN ATTRIBUTE RECOGNITION

Early image-based PAR approaches primarily treated the task as a multi-label classification problem using convolutional neural networks (CNNs). Abdulnabi et al. [19] proposed a multi-task learning framework that employed multiple CNNs to learn attribute-specific features while sharing knowledge across networks. (Zhang et al., 2014) introduced PANDA, integrating a part-aware model with CNN-based attribute classification to learn robust normalized features even from smaller datasets. With the recognition that pedestrian attributes exhibit strong correlations, researchers began exploring sequential modeling approaches. More recent approaches have focused on attention mechanisms and region localization. (Wu et al., 2024) aimed to capture discriminative features from regions that are easily overlooked. (Zhou et al., 2024) introduced a probabilistic technique that help to enhance data using a Bayesian feature augmentation method. PromptPAR (Wang et al., 2025a) jointly modeled attribute context and image-attribute relationships. (Setyawan et al., 2024) introduced PARformer, a Transformer-based approach combining global and local perspectives.VTB (Cheng et al., 2022) treated PAR as a vision-text interaction problem by introducing a text encoder to enhance feature representation. Video-based PAR represents a newer research direction that leverages temporal information across video frames. (Wang et al., 2024) adopted CLIP (Radford et al., 2021) to extract the visual features.

### 2.2 MIXTURE OF EXPERTS

Mixture of Experts (MoE) models have emerged as an efficient approach to scale model capacity without proportionally increasing computational costs. In MoE architectures, only a subset of parameters is activated for each input token, enabling models to grow in parameter count while maintaining reasonable inference and training costs. MoE Transformers typically replace standard feed-forward networks with expert layers where each token is dynamically routed to a small subset of available experts. This sparse activation strategy allows models to scale to billions or trillions of parameters while keeping computation manageable.

Various routing mechanisms have been proposed to determine expert-token assignments. The greedy top-$k$ experts per token approach (Shazeer et al., 2017) selects the $k$ experts with the highest routing scores for each input token, allowing tokens to benefit from multiple specialized networks simultaneously. In contrast, the greedy top-$k$ tokens per expert mechanism (Zhou et al., 2022) prioritizes

each expert's perspective by allowing experts to select tokens they are most confident in processing, potentially improving specialization and efficiency.

A key challenge in MoE training is load balancing. Without proper constraints, tokens may disproportionately route to certain experts, creating training inefficiencies. To address this, auxiliary losses are often employed to promote balanced expert utilization and minimize unassigned tokens. Successful MoE implementations include GShard (Lepikhin et al., 2021) and Switch Transformers (Fedus et al., 2022) for language tasks, and V-MoE (Riquelme et al., 2021) for vision tasks. These approaches demonstrate that MoE architectures can achieve superior performance compared to dense models with equivalent computational budgets. (Riquelme et al., 2021) explore the effectiveness of MoE in scaling vision-language models. As illustrated in Figure 1, conventional Top-$k$ routers typically rely on visual features alone to determine expert assignment. While effective in standard recognition settings, this design overlooks semantic cues that are highly informative in attribute recognition tasks. In contrast, our method differs in both routing and expert design. VLAAR's router jointly uses image features and language-derived attribute embeddings, so experts are selected by semantic relevance to attributes rather than visual similarity alone. Moreover, instead of a homogeneous expert set as in standard MoE, VLAAR employs a heterogeneous pool of Standard, Advanced (residual MLP), and Convolutional experts, better matching the diverse nature of pedestrian attributes and yielding gains over a uniform expert pool in our ablations.

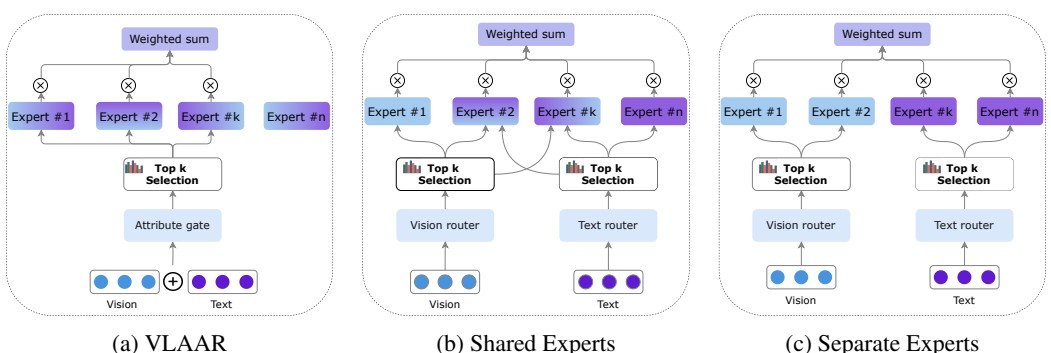

(a) VLAAR      (b) Shared Experts      (c) Separate Experts

Figure 1: **Comparison of Top-$k$ router designs.** (a) VLAAR combines visual and text embeddings through an attribute gate to guide expert selection; (b) a shared-expert design uses separate vision and text routers but assigns both modalities to the same expert pool; (c) a separate-expert design routes each modality to its own tailored expert pool.

## 3 METHOD

This section details our parameter-efficient mixture-of-experts architecture for multi-label video-based pedestrian attribute recognition. The overall pipeline is shown in Figure 2. Our approach leverages a frozen, pre-trained CLIP complemented by lightweight expert modules that are dynamically activated through a novel dual-modality routing mechanism. This mechanism seamlessly integrates visual representations with semantic attribute priors, enabling sophisticated task-specific feature refinement while preserving valuable pre-trained knowledge.

### 3.1 OVERALL ARCHITECTURE

Our architecture comprises three synergistic components operating within an end-to-end framework. First, a frozen pre-trained visual backbone CLIP(ViT-B/16) functions as the feature extractor, transforming input video frames $\mathbf{V} \in \mathbb{R}^{T \times H \times W \times 3}$ into rich hierarchical representations $\mathbf{X} \in \mathbb{R}^{T \times d}$, where $T$ denotes the number of frames, $H$ and $W$ represent spatial dimensions, and $d$ is the feature dimension.

Second, VLAAR processes the extracted visual features by fusing them with semantic attribute cues derived from natural language prompts $\{p_g\}_{g=1}^{G}$, where $G$ represents the number of attribute groups. This router serves as an intelligent gatekeeper, analyzing both visual and textual modalities to determine the optimal expert selection strategy for each input instance.

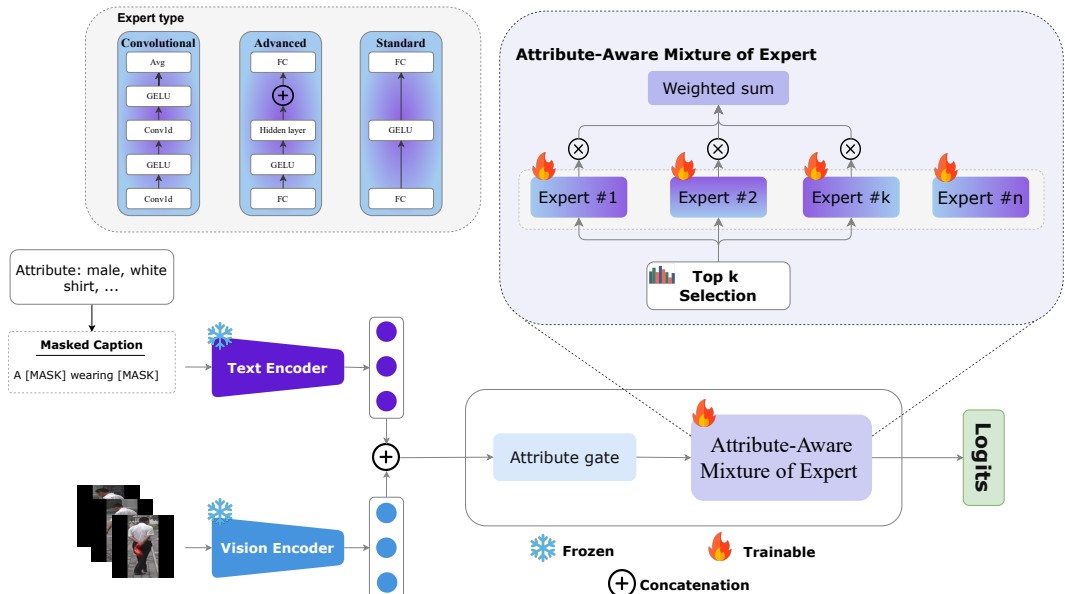

Figure 2: **Overview of the proposed pipeline.** The CLIP backbone encodes the input video $\mathbf{V} \in \mathbb{R}^{T \times H \times W \times 3}$ into hierarchical visual features. An attribute-aware mixture-of-experts (MoE), routed by vision–language signals, selects a sparse subset of experts to refine these features for multi-label attribute prediction. Experts are drawn from three families: *Convolutional*, *Advanced*, and *Standard*.

Third, based on the router's decisions, our Sparse Mixture-of-Experts (SMoE) system dynamically activates the most relevant subset of expert modules $\{\text{Expert}_i\}_{i=1}^n$. Each expert specializes in recognizing particular attribute patterns (e.g., clothing style, accessories, age groups), and they are strategically integrated to inject task-specific information while maintaining the structural integrity of the pre-trained features. The selected experts collectively form a comprehensive attribute recognition system that processes the routed features to generate accurate attribute predictions $\hat{\mathbf{y}} \in \mathbb{R}^C$, where $C$ denotes the total number of attributes.

## 3.2 VISION-LANGUAGE ATTRIBUTE-AWARE ROUTER (VLAAR)

A key innovation in our approach is the Vision-Language Attribute-Aware Router (VLAAR), which integrates semantic information via natural language prompts corresponding to different attribute categories. For each attribute group $g$ (e.g., "The person is wearing [ATTRIBUTE]", "The person has [ATTRIBUTE]"), we define a corresponding prompt template encoded using a frozen text encoder: $\mathbf{p}_g = E_{\text{text}}(\text{prompt}_g) \in \mathbb{R}^d$, where $E_{\text{text}}$ is a pre-trained text encoder (CLIP). Different prompt design strategies and their comparative analysis are presented in Appendix A.1. These language embeddings inject rich semantic context into the routing process. The combined routing logit vector is obtained by integrating the visual features $\mathbf{x} \in \mathbb{R}^d$ from the vision backbone with a weighted sum of the language embeddings $\mathbf{p}_g$:

$$\mathbf{z} = \alpha \, \mathbf{W}_v \, \mathbf{x} + (1 - \alpha) \, \mathbf{W}_l \left( \sum_{g=1}^{G} \beta_g \, \mathbf{p}_g \right) \tag{1}$$

where $\alpha \in [0, 1]$ is a learnable parameter balancing visual and linguistic information, $\mathbf{W}_v \in \mathbb{R}^{n \times d}$ and $\mathbf{W}_l \in \mathbb{R}^{n \times d}$ are projection matrices, $\beta_g$ represents the relevance of each attribute group to the current input is computed using a scaled dot-product attention mechanism:

$$\beta_g = \frac{\exp\left( \frac{(\mathbf{W}_q \, \mathbf{x})^\top (\mathbf{W}_k \, \mathbf{p}_g)}{\sqrt{d}} \right)}{\sum_{j=1}^{G} \exp\left( \frac{(\mathbf{W}_q \, \mathbf{x})^\top (\mathbf{W}_k \, \mathbf{p}_j)}{\sqrt{d}} \right)} \tag{2}$$

where $\mathbf{W}_q \in \mathbb{R}^{d \times d'}$ and $\mathbf{W}_k \in \mathbb{R}^{d \times d'}$ transform features into a common space with dimension $d'$.

After computing the routing logits, we apply a sparse softmax operation to obtain the final gating vector $G(\mathbf{x}, \{\mathbf{p}_g\}_{g=1}^G) = \text{SparseSoftmaxTopK}(\mathbf{z})$. The complete output representation is:

$$\mathbf{y} = \mathbf{x} + \sum_{i=1}^{n} \left[ G(\mathbf{x}, \{\mathbf{p}_g\}_{g=1}^G) \right]_i \text{Expert}_i(\mathbf{x}) \tag{3}$$

### 3.3 Attribute-Aware Mixture of Expert

#### 3.3.1 Expert Module Structure

To effectively handle the diverse nature of pedestrian attributes, we employ three distinct expert architectures, each designed to capture different types of feature relationships. For the visual features $\mathbf{x} \in \mathbb{R}^d$ extracted from the backbone, each expert applies specialized transformations based on its architectural design, enabling the model to adaptively select the most appropriate computational pattern for different attribute types. Our expert pool comprises three architectural variants:

**Standard Expert** follows a bottleneck design:

$$\text{StandardExpert}_i(\mathbf{x}) = \text{Dropout}\big(\text{FC}_{\text{up}}\big(\sigma\big(\text{Norm}\big(\text{FC}_{\text{down}}(\mathbf{x})\big)\big)\big)\big) \tag{4}$$

**Advanced Expert** employs multi-layer architectures with residual connections:

$$\text{AdvancedExpert}_i(\mathbf{x}) = \text{LayerNorm}(HiddenLayer + \text{Residual}(\mathbf{x})) \tag{5}$$

where $\mathbf{h}_L$ represents the output after $L$ sequential transformations with intermediate residual connections.

**Convolutional Expert** utilizes 1D convolutions to capture local feature patterns:

$$\text{ConvExpert}_i(\mathbf{x}) = \text{FC}(\text{Pool}(\text{Conv1D}(\text{reshape}(\mathbf{x})))) \tag{6}$$

The architectural diversity of our expert pool enables complementary specialization for heterogeneous pedestrian attributes. Standard experts perform parameter-efficient linear transformations, well-suited for simple binary attributes. Advanced experts with residual connections capture complex inter-attribute dependencies that require multi-level reasoning. Convolutional experts use 1D convolutions to exploit spatial structure preserved in the feature representations, which is particularly important for region-specific appearance cues. Our vision-language routing mechanism leverages semantic attribute information to dynamically select among these expert types: simple attributes tend to rely on standard experts, relational attributes on advanced experts, and spatially localized attributes on convolutional experts. As confirmed by our ablation studies, this division of labor allows the model to allocate computational capacity according to both visual complexity and semantic attribute characteristics, making the three expert types complementary rather than redundant.

#### 3.3.2 Sparse Mixture-of-Experts (SMoE)

For an input feature vector $\mathbf{x} \in \mathbb{R}^d$, our SMoE layer selectively activates only a subset of experts to process each input, enhancing both computational efficiency and specialization. Our expert pool consists of $n_s$ standard experts, $n_a$ advanced experts, and $n_c$ convolutional experts, totaling $n = n_s + n_a + n_c$ experts. This heterogeneous composition allows the routing mechanism to select not only which experts to activate but also which computational patterns are most suitable for processing different attribute types. The output of the SMoE is computed as:

$$\text{SMoE}(\mathbf{x}) = \sum_{i=1}^{n} g_i(\mathbf{x}) \text{Expert}_i(\mathbf{x}) \tag{7}$$

where $g_i(\mathbf{x})$ represents the gating weight assigned to expert $i$ for input $\mathbf{x}$, considering both the expert's relevance and its architectural suitability for the current attribute recognition task.

The gating mechanism begins by computing routing logits $z_i = [\mathbf{W}\mathbf{x}]_i$ using a learnable routing matrix $\mathbf{W} \in \mathbb{R}^{n \times d}$, where $n$ is the number of experts. Rather than activating all experts, we identify the Top-$k$ experts with the highest routing scores for each input, denoted as $\mathcal{S}_K(\mathbf{x}) = \text{TopK}(z_1, z_2, \ldots, z_n)$. We then normalize the weights of only the selected experts using a softmax function applied exclusively to the Top-$k$ experts:

$$g_i(\mathbf{x}) = \begin{cases} \frac{\exp(z_i)}{\sum_{j \in \mathcal{S}_K(\mathbf{x})} \exp(z_j)} & \text{if } i \in \mathcal{S}_K(\mathbf{x}), \\ 0 & \text{otherwise.} \end{cases} \tag{8}$$

This formulation ensures that only K out of n experts are activated for each input, with the activated experts receiving normalized weights that sum to 1, while non-selected experts receive zero weight and are effectively excluded from computation. The sparse activation reduces computational requirements by approximately a factor of $\frac{n}{K}$ compared to dense activation of all experts, providing an efficient mechanism for specialized feature processing.

### 3.4 LOSS FUNCTION AND TRAINING OBJECTIVE

We formulate the problem as a multi-label classification task with an integrated loss function combining binary cross-entropy with auxiliary components for load balancing:

$$\mathcal{L} = \underbrace{-\frac{1}{C} \sum_{i=1}^{C} [y_i \log(\hat{y}_i) + (1 - y_i) \log(1 - \hat{y}_i)]}_{\mathcal{L}_{\text{BCE}}} + \lambda_{\text{aux}} \mathcal{L}_{\text{aux}} \tag{9}$$

where $y_i \in 0, 1$ is the ground-truth label and $\hat{y}_i \in [0, 1]$ is the predicted probability of the $i$-th attribute (out of $C$ attributes), obtained by projecting the weighted expert outputs $\mathbf{y}$. We apply label smoothing with factor $\epsilon = 0.1$ to prevent overconfident predictions and improve generalization.

The auxiliary loss incorporates load balancing components to prevent expert collapse: $\mathcal{L}_{\text{aux}} = \mathcal{L}_{\text{freq}} + \lambda_{\text{imp}} \mathcal{L}_{\text{imp}}$, where $\lambda_{\text{imp}} = 0.1$ is a weighting hyperparameter. The frequency and importance components are defined as:

$$\mathcal{L}_{\text{freq}} = \sum_{i=1}^{n} P_i \log(P_i) \cdot n$$

$$\mathcal{L}_{\text{imp}} = \sum_{i=1}^{n} \left( \frac{1}{B \cdot T} \sum_{b=1}^{B} \sum_{t=1}^{T} \mathbb{I}[i \in \mathcal{S}_K^{b,t}] \cdot G(\mathbf{x}^{b,t}, \{\mathbf{p}_g\}_{g=1}^{G})_i \right)^2 \cdot n \tag{10}$$

where $P_i = \frac{1}{B \cdot T} \sum_{b=1}^{B} \sum_{t=1}^{T} \mathbb{I}[i \in \mathcal{S}_K^{b,t}]$ represents the fraction of tokens routed to expert $i$ across a batch of $B$ samples, each containing $T$ tokens. This load balancing encourages all experts to receive comparable routing probability while specializing in different regions of the attribute space, ensuring accurate attribute recognition and efficient expert utilization.

## 4 EXPERIMENTS

In this section, we present our experimental setup and results. We first introduce our datasets and evaluation metrics in Section 4.1, followed by implementation details in Section 4.2. Section 4.3 compares our method with state-of-the-art approaches and Section 4.4 compares our method with other PEFT methods. Finally, in Section 4.5 Ablation studies are performed to evaluate how modules contributes to VLAAR's performance and the trade off for choosing the hyper-parameters.

### 4.1 DATASETS AND EVALUATION METRICS

**Image Benchmarks** We evaluate on three standard datasets: PETA (Deng et al., 2014) (19,000 images, 35 attributes), PA100K (Liu et al., 2017) (100,000 images, 26 attributes), and RAPv1 (Li et al., 2019) (41,585 images, 51 attributes).

Table 1: **Comparison with state-of-the-art methods** on the image-based datasets. Best results are in bold, and second-best are underlined. "–" means the value is unavailable in the original paper.

| Method | PETA | | | | | PA100K | | | | | RAPv1 | | | | |
|---|---|---|---|---|---|---|---|---|---|---|---|---|---|---|---|
| | mA | Acc | Prec | Rec | F1 | mA | Acc | Prec | Rec | F1 | mA | Acc | Prec | Rec | F1 |
| IAA (Wu et al., 2021) | 85.27 | 78.04 | 86.08 | 85.80 | 85.64 | 81.94 | 80.31 | 88.36 | 88.01 | 87.80 | 81.72 | 68.47 | 79.56 | 82.06 | 80.37 |
| DRFormer (Tang & Huang, 2022) | 89.96 | 81.30 | 85.68 | 91.08 | 88.30 | 82.47 | 80.27 | 87.60 | 88.49 | 88.04 | 81.81 | 70.60 | 80.12 | 82.77 | 81.42 |
| VAC (Guo et al., 2022) | – | – | – | – | – | 82.19 | 80.66 | 88.72 | 88.10 | 88.41 | 81.30 | 70.12 | 81.56 | 81.51 | 81.54 |
| DAFL (Jia et al., 2022) | 87.07 | 78.88 | 85.78 | 87.03 | 86.40 | 83.54 | 80.13 | 87.01 | 89.19 | 88.09 | 83.72 | 68.18 | 77.41 | 83.39 | 80.29 |
| VTB (Cheng et al., 2022) | 85.31 | 79.60 | 86.76 | 87.17 | 86.71 | 83.72 | 80.89 | 87.88 | 89.30 | 88.21 | 82.67 | 69.44 | 78.28 | 84.39 | 80.84 |
| PARformer (Fan et al., 2024) | 89.32 | 82.86 | 88.06 | 91.98 | 89.06 | 84.46 | 81.13 | 88.09 | 91.67 | 88.52 | 84.43 | 69.94 | 79.63 | 88.19 | 81.35 |
| Solider (Chen et al., 2023) | 85.12 | 87.84 | 88.27 | 84.21 | 86.32 | 86.61 | 89.99 | 87.12 | 84.21 | 89.67 | - | - | - | - | - |
| SOFA (Wu et al., 2024) | 87.10 | 81.10 | 87.80 | 88.40 | 87.80 | 83.40 | 81.10 | 88.40 | 89.00 | 88.30 | 83.40 | 70.00 | 80.00 | 83.00 | 81.20 |
| FRDL (Zhou et al., 2024) | 88.59 | – | – | – | 89.03 | 89.44 | – | – | – | 88.05 | 87.72 | - | - | - | 79.16 |
| PLIP (Zuo et al., 2024) | 87.12 | 89.52 | 88.57 | 84.00 | 88.84 | 89.26 | 83.99 | 88.21 | 85.20 | 90.75 | - | - | - | - | - |
| Hulk (Wang et al., 2025b) | 85.20 | 82.15 | 86.80 | 87.20 | 87.00 | 87.85 | 86.30 | 87.95 | 82.80 | 85.26 | 82.50 | 68.90 | 79.20 | 81.20 | 80.15 |
| EVSITP (Wu et al., 2025) | 89.65 | 83.93 | 89.67 | 90.73 | 90.20 | 88.66 | 84.54 | 89.90 | 92.09 | 90.98 | 86.10 | 71.64 | 79.24 | 86.65 | 82.78 |
| VLAAR (Ours) | 90.12±0.06 | 90.45±0.06 | 89.32±0.06 | 93.15±0.07 | 91.22±0.07 | 90.88±0.07 | 91.47±0.08 | 90.23±0.08 | 91.32±0.08 | 92.18±0.07 | 86.35±0.06 | 72.28±0.06 | 83.74±0.07 | 89.65±0.06 | 83.42±0.07 |

Table 2: **Comparison with state-of-the-art methods on the video-based datasets**. We report the results on MARS and DukeMTMC-VID, along with their **Average**. Best results are in bold, and second-best are underlined. "–" means the value is unavailable.

| Method | Backbone | MARS-Attribute | | | | DukeMTMC-VID-Attribute | | | | Average | | | |
|---|---|---|---|---|---|---|---|---|---|---|---|---|---|
| | | Acc | Prec | Rec | F1 | Acc | Prec | Rec | F1 | Acc | Prec | Rec | F1 |
| TA(Image) (Chen et al., 2019) | ResNet50 | 85.85 | – | – | 67.28 | 87.77 | – | – | 68.70 | 86.81 | – | – | 67.99 |
| TA(Video) (Chen et al., 2019) | ResNet50 | 87.01 | – | – | 72.04 | 89.31 | – | – | 73.24 | 88.16 | – | – | 72.64 |
| Lee et al. (Lee et al., 2022) | ResNet50 | 86.75 | – | – | 70.42 | 88.98 | – | – | 72.30 | 87.87 | – | – | 71.36 |
| TRA (Zhao et al., 2023) | ResNet50 | 87.05 | – | – | 71.92 | 89.32 | – | – | 75.01 | 88.19 | – | – | 73.47 |
| VTB (Cheng et al., 2022) | ViT-B/16 | 90.37 | 78.96 | 78.42 | 78.32 | 90.29 | 73.38 | 76.99 | 74.81 | 90.33 | 76.17 | 77.71 | 76.57 |
| VTF (Zhu et al., 2024) | ViT-B/16 | 92.47 | 81.76 | 82.95 | 81.94 | 92.45 | 77.23 | – | 78.83 | 92.46 | 79.50 | – | 80.39 |
| VTF++ (Wang et al., 2024) | ViT-B/16 | 93.19 | 82.27 | 84.87 | 83.22 | 93.31 | 78.19 | 84.80 | 80.45 | 93.25 | 80.23 | 84.84 | 81.84 |
| VLAAR (Ours) | ViT-B/16 | 94.73±0.04 | 86.21±0.05 | 87.45±0.07 | 87.89±0.08 | 94.56±0.05 | 82.84±0.07 | 88.73±0.09 | 84.28±0.08 | 94.65±0.04 | 84.53±0.06 | 88.09±0.08 | 86.09±0.07 |

**Video Benchmarks** For video-based evaluation, we use MARS-Attribute (Zheng et al., 2016) (17,503 sequences from 1,261 identities with 12 attributes including gender, age, and clothing) and DukeMTMC-VID-Attribute (Wu et al., 2018) (4,832 tracklets from 1,404 identities with 16 binary attributes). Both datasets capture multi-camera surveillance with sequences ranging from 2 to 920 frames. Each video is modeled as a temporal sequence: the backbone extracts hierarchical representations, and the VLAAR router and MoEs operate directly on these token representations, dynamically assigning experts based on the visual and textual context.

## 4.2 IMPLEMENTATION DETAILS

We configure our expert pool with 2 standard experts, 2 advanced experts (with 3 hidden layers each), and 2 convolutional experts, totaling 6 experts. Detailed analysis of hyperparameter selection, including the choice of K=2 and 6 experts, is provided in Appendix A.2. The advanced experts use residual connections and layer normalization, while convolutional experts employ 1D convolutions with kernel size 3 and adaptive average pooling. We train the model on 1xA100 using the Adam optimizer (Kingma & Ba, 2015) with $\beta_1 = 0.9$, $\beta_2 = 0.999$, and a weight decay of $1 \times 10^{-5}$. The initial learning rate is set to $3 \times 10^{-4}$ with a cosine annealing schedule and 5-epoch linear warmup. Top k value chosen is 2 and number of experts is 6. For the loss hyperparameters, we set $\lambda_{aux} = 0.1$ based on validation performance. During inference, we apply a threshold of 0.5 to the sigmoid outputs to determine the presence of each attribute.

## 4.3 RESULTS ON IMAGE AND VIDEO BENCHMARKS

We conduct extensive experiments on both image-based and video-based pedestrian attribute recognition benchmarks to evaluate the effectiveness of our proposed approach.

**Image-based Benchmark Results** Table 1 presents comprehensive comparisons on three widely-adopted image-based PAR benchmarks. On PETA, VLAAR achieves state-of-the-art performance with 90.12% mA, 90.45% accuracy, 89.32% precision, 93.15% recall, and 91.22% F1 score, substantially outperforming previous methods across all metrics. For PA100K, our method delivers 90.88% mA, 91.47% accuracy, 90.23% precision, 91.32% recall, and 92.18% F1 score, representing significant improvements over existing approaches. On the challenging RAPv1 dataset, VLAAR obtains 86.35% mA, 72.28% accuracy, 83.74% precision, 89.65% recall, and 83.42% F1 score. These consistent improvements across diverse datasets validate the effectiveness of our vision-language routing mechanism and heterogeneous expert design.

Table 3: Comparison of Trainable Parameters, FLOPs, and Inference Time.

| Method | #Trainable Param. (M) | FLOPs (G) | Inference Time (ms) |
|---|---|---|---|
| PARformer (Fan et al., 2024) | 84.04 | 6.5 | 28.45 |
| Solider (Chen et al., 2023) | 48.41 | **4.0** | 27.12 |
| SOFA (Zhou et al., 2024) | 86.55 | 6.7 | 29.34 |
| FRDL (Zhou et al., 2024) | 81.93 | 5.3 | 28.98 |
| PLIP (Zuo et al., 2024) | 79.76 | 5.2 | 27.76 |
| Hulk (Wang et al., 2025b) | 73.28 | 4.8 | 26.89 |
| EVSITP (Wu et al., 2025) | 76.41 | 5.0 | 27.33 |
| **VLAAR (Ours)** | **26.92** | 4.2 | **24.18** |

**Video-based Benchmark Results**   We evaluate our approach on two standard video-based pedestrian attribute recognition datasets, as shown in Table 2. On MARS-Attribute, VLAAR achieves 94.73% accuracy, 86.21% precision, 87.45% recall, and 87.89% F1 score, surpassing the previous state-of-the-art VTF++ by 1.54% in accuracy and 4.67% in F1 score. For DukeMTMC-VID-Attribute, our method attains 94.56% accuracy, 82.84% precision, 88.73% recall, and 84.28% F1 score, demonstrating improvements of 1.25% and 4.67% in accuracy and F1 score respectively over VTF++. These substantial gains confirm that our temporal modeling strategy and attribute-aware feature learning mechanism effectively leverage video information for enhanced attribute recognition.

Table 3 summarizes trainable parameters and inference latency for recent pedestrian attribute recognition models. After retuning, prior methods operate within a comparable latency band that meets our deployment constraints. *VLAAR* remains the fastest while also using the fewest trainable parameters.

## 4.4 RESULTS WITH PARAMETER-EFFICIENT FINE-TUNING

To assess the effectiveness of VLAAR, we compare it with recent PEFT methods (LoRA, Prompt-Tuning, Adapter-Tuning, and VTF-par++). Table 4 shows results on the MARS-Attribute and DukeMTMC-VID-Attribute datasets. VLAAR consistently outperforms all baselines, achieving F1 score gains of 4.67% and 3.83% over the previous state-of-the-art (VTF-par++). Although VLAAR uses 26.92M trainable parameters—more than LoRA (7.54M) and Prompt-Tuning (7.10M)—it maintains competitive inference time (24.18ms) while delivering substantially higher accuracy. These results confirm that attribute-aware routing enables more effective feature adaptation and demonstrate a strong performance–efficiency trade-off suitable for real-world deployment.

Table 4: **Comparison with PEFT methods on the video-based datasets**. All methods employ ViT-B/16 as the backbone.

| Method | # Trainable Param. (M) | Inference Time (ms) | MARS-Attribute | | DukeMTMC-VID-Attribute | |
|---|---|---|---|---|---|---|
| | | | Acc | F1 | Acc | F1 |
| LoRA (Hu et al., 2022) | 7.54 | **18.3** | 91.23 | 82.45 | 91.32 | 77.69 |
| Prompt-Tuning (Lester et al., 2021) | 7.10 | 17.9 | 89.57 | 81.88 | 89.64 | 74.93 |
| Adapter-Tuning (Houlsby et al., 2019) | 12.34 | 23.5 | 92.14 | 82.58 | 91.86 | 79.72 |
| VTF-PAR++ (Wang et al., 2024) | 15.04 | 24.7 | 93.19 | 83.22 | 93.31 | 80.45 |
| **VLAAR (k = 2, #experts = 6)** | **26.92** | 24.8 | **94.73** | **87.89** | **94.56** | **84.28** |

## 4.5 ABLATION STUDY

### 4.5.1 EXPERT ARCHITECTURE DIVERSITY ANALYSIS

To validate the benefit of using diverse expert architectures, we compare our heterogeneous expert design against homogeneous alternatives. Table 5 shows that combining different expert types yields superior performance compared to using only one type of expert.

The mixed configuration demonstrates that architectural diversity enables more effective specialization for different attribute recognition patterns. We compare three different Top-$k$ router architectures (Figure 1) to validate our VLAAR design. As shown in Table 6, the VLAAR joint router consistently outperforms alternative designs on both datasets, achieving F1 score improvements of up to

1.62 percentage points. These results confirm that VLAAR's vision-language fusion before routing enables more effective expert selection compared to processing modalities independently, providing an optimal balance between computational efficiency and attribute recognition performance.

Table 5: Ablation study of expert architecture configurations on the MARS-Attribute dataset.

| Expert Configuration | Acc | F1 |
|---|---|---|
| Standard Experts | 92.84 | 84.12 |
| Advanced Experts | 93.01 | 84.58 |
| Convolutional Experts | 92.76 | 83.89 |
| **VLAAR** | **94.73** | **87.89** |

Table 6: Ablation study of different Top-$k$ router designs on video-based benchmarks. All variants use ViT-B/16 as the backbone with 6 experts and $k = 2$. Reported numbers correspond to F1.

| Router Design | MARS-Attribute | DukeMTMC-VID-Attribute |
|---|---|---|
| **VLAAR (Joint Router)** | **87.89** | **84.28** |
| Modality-Specific (Shared Experts) | 84.68 | 81.54 |
| Modality-Specific (Separate Experts) | 84.89 | 81.78 |

## 5 CONCLUSION

This paper introduces VLAAR, a novel parameter-efficient framework for pedestrian attribute recognition that leverages a sparse mixture-of-experts architecture with vision-language guided routing. Our dual-modality routing mechanism dynamically integrates visual features with semantic attribute cues from natural language prompts, enabling more effective expert selection for fine-grained attribute recognition. Experimental evaluations on multiple benchmarks demonstrate that VLAAR achieves state-of-the-art performance on both image-based and video-based datasets, with significant improvements in accuracy and F1 scores while maintaining parameter efficiency. Our ablation studies reveal that the joint vision-language router consistently outperforms modality-specific alternatives. The VLAAR framework demonstrates that carefully designed sparse expert architectures with vision-language fusion can substantially enhance pedestrian attribute recognition capabilities while preserving computational efficiency, making it suitable for practical deployment in surveillance and person re-identification systems.

## 6 LIMITATIONS

**Computational Overhead During Training**. One of the key aspects of our VLAAR framework is dynamically routing input features to specialized expert modules. This creates significant memory demands during training, as all expert parameters must be maintained in memory even though only a subset is activated for each input. Although our sparse activation strategy reduces computation during inference, the training process still requires allocating memory for all experts simultaneously.

**Cross-Dataset Generalization**. Our experiments primarily evaluated performance on within-dataset scenarios. The generalization capability across datasets with different camera viewpoints, lighting conditions, and attribute distributions remains an open question that deserves further investigation.

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
