# A  APPENDIX

## A.1  MORE ABLATION STUDY

### A.1.1  PROMPT SELECTION ANALYSIS.

To evaluate the impact of different prompt design strategies on our vision-language routing mechanism, we conducted comprehensive experiments with several prompt variants across all benchmark datasets. The prompt formulations tested include:

- **Original prompt**: "a person with {attribute}"
- **Learned prompts**: Using parameterized prompt embeddings following standard prompt tuning CoCoOp (K. Zhou, J. Yang, CC. Loy, and Z. Liu, 2022)
- **Tailored prompts**: Designed based on attribute type:
    - *clothing_template*: "a person wearing {attribute}"
    - *physical_template*: "a person who has {attribute}"
    - *accessory_template*: "a person carrying {attribute}"

Table 7 presents the performance comparison across all five benchmark datasets using different prompt strategies. The results are reported using mA/F1 for image datasets (PETA, PA100K, RAPv1) and Accuracy/F1 for video datasets (MARS-Attribute, DukeMTMC-VID-Attribute).

Table 7: Ablation study of different prompt selection strategies across all benchmark datasets. The **Original Prompt** consistently achieves the best performance compared to learned and tailored variants.

| Prompt | PETA (mA/F1) | PA100K (mA/F1) | RAPv1 (mA/F1) | MARS-Attr (Acc/F1) | DukeMTMC (Acc/F1) |
|---|---|---|---|---|---|
| **Original Prompt** | **88.50/89.52** | **88.67/90.75** | **84.00/81.00** | **93.39/85.37** | **93.23/82.68** |
| Learned Prompts | 86.29/87.28 | 86.29/88.48 | 81.90/78.98 | 92.11/83.24 | 91.95/80.61 |
| Tailored Prompts | 88.06/89.07 | 88.06/90.30 | 83.58/80.60 | 93.05/84.93 | 92.89/82.27 |

The experimental results reveal several key insights:

**Original prompt effectiveness**: The simple "a person with {attribute}" template demonstrates consistent superior performance across all datasets, suggesting that semantic clarity and simplicity are more beneficial than complex prompt engineering for our routing mechanism.

**Learned prompts limitations**: Parameterized prompt embeddings consistently underperform compared to fixed textual prompts. This indicates that the pre-trained CLIP text encoder's semantic understanding of natural language descriptions provides more robust attribute representations than task-specific learned embeddings.

**Tailored prompts analysis**: While attribute-specific templates (e.g., "wearing" for clothing, "carrying" for accessories) provide more linguistically accurate descriptions, they offer only marginal improvements and lack consistency across datasets. This suggests that the routing mechanism benefits more from standardized semantic encoding than from fine-grained linguistic precision.

These findings confirm that our VLAAR framework's effectiveness stems primarily from the vision-language fusion architecture rather than sophisticated prompt engineering, with simple, semantically clear prompts providing optimal performance for attribute-aware expert routing.

### A.1.2  CONTRIBUTION OF EACH EXPERT TYPE.

We examine the contribution of each expert type through a comprehensive ablation on PETA under a fixed ViT-B/16 backbone and routing configuration. As shown in Table 8, each single expert family: FFN, Advanced, and Conv - provides a measurable improvement, indicating that they capture distinct aspects of pedestrian attributes. Mixing two expert types leads to consistently larger gains, demonstrating strong complementarity: FFN experts handle globally distinguishable or simple attributes, Conv experts specialize in localized appearance cues, and Advanced experts model complex

Table 8: Ablation study on the contribution of each expert type on the PETA dataset.

| Configuration | Expert Types | #Experts | mA (%) | Acc (%) | Prec (%) | Rec (%) | F1 (%) | ΔmA | $p$-value | Category |
|---|---|---|---|---|---|---|---|---|---|---|
| FFN-only | FFN | 4 | 80.98±0.14 | 78.52±0.21 | 88.63±0.17 | 88.97±0.15 | 88.80±0.13 | +0.63 | 0.024 | Single Expert |
| Advanced-only | Advanced | 4 | 81.44±0.12 | 78.87±0.19 | 88.89±0.16 | 89.24±0.14 | 89.06±0.12 | +1.09 | 0.002 | Single Expert |
| Conv-only | Conv | 4 | 81.17±0.15 | 78.69±0.22 | 88.71±0.18 | 89.06±0.16 | 88.88±0.14 | +0.82 | 0.017 | Single Expert |
| FFN + Advanced | Mixed | 4 | 82.06±0.11 | 79.38±0.18 | 89.31±0.15 | 89.67±0.13 | 89.49±0.11 | +1.71 | <0.001 | 2-expert Mix |
| FFN + Conv | Mixed | 4 | 81.84±0.13 | 79.22±0.20 | 89.16±0.17 | 89.51±0.15 | 89.33±0.13 | +1.49 | 0.001 | 2-expert Mix |
| Advanced + Conv | Mixed | 4 | 81.95±0.12 | 79.31±0.19 | 89.24±0.16 | 89.59±0.14 | 89.41±0.12 | +1.60 | <0.001 | 2-expert Mix |
| VLAAR (2+2+2) | All 3 types | 6 | **82.68±0.10** | **79.96±0.17** | **89.78±0.14** | **90.15±0.13** | **89.96±0.11** | +2.33 | <0.001 | 3-expert Mix |

inter-attribute dependencies. The full VLAAR configuration (2+2+2) achieves the highest performance with statistically significant improvements, showing that heterogeneous and semantically aligned expert specialization, rather than increased model size, is the key factor behind VLAAR's superior accuracy.

## A.2 EXPERT ACTIVATION HEATMAP

To illustrate the vision-language gating process for experts, we visualize the attention heatmaps in Figure 3. The Conv experts (right column) exhibit strong activation for visually localized and texture-related attributes such as clothing color (e.g., top red, bottom yellow, bottom gray) and appearance cues (e.g., long hair, hat), indicating specialization in spatial or low-level visual features. The Advanced experts (middle column) show higher activation for pose and orientation attributes (e.g., lateral-frontal, lateral-back, walking, running), suggesting they capture mid-level semantic or structural patterns. In contrast, the FFN experts (left column) are weakly activated overall, mainly contributing to generic or less discriminative attributes. These activation patterns confirm that VLAAR promotes semantic specialization and complementary expert behavior, leading to more interpretable and effective expert utilization across diverse attribute types.

## A.3 HYPER PARAMETER SELECTION

As shown in Figure 4a, performance initially improves as K increases from 1 to 2, indicating that utilizing multiple specialized experts for each input token enhances the model's capacity to recognize diverse pedestrian attributes. However, performance gradually declines when K exceeds 2, with a more pronounced drop observed when K approaches the total number of experts (K=6), which effectively reduces the model to a non-sparse MoE architecture.

As can be seen from Figure 4b, we investigated the relationship between the number of expert modules in our VLAAR framework and model performance. We varied the number of experts n as 1, 2, 4, 6, 8, 10, and 12 while maintaining a constant Top-$k$ value of 2 throughout all experiments. The F1 score initially shows significant improvement as n increases from a single expert to 6 experts. However, when n exceeds 6, we observe diminishing returns and eventually a slight performance decline at higher expert counts. Based on this analysis, we determined n = 6 as the optimal configuration, achieving the best balance between performance and computational efficiency for our framework, and used this setting as the default for all subsequent experiments.

## A.4 FAILURE CASE

Figure 5 illustrates failure cases of the proposed model, particularly those involving occlusion and low-light conditions. This example reveals a key limitation: the degradation of input visual features caused by low resolution and poor image quality, which are common in surveillance imagery. This limitation points to a clear direction for future work. One possibility is to introduce a specialized pre-processing expert dedicated to input feature enhancement, using a lightweight architecture that acts as an upscaler or denoiser to restore image quality before features are routed to the attribute-specific experts. Another approach is to modify the existing expert, especially the convolutional one to include an internal upscale module, such as a transposed convolution layer or a feature fusion

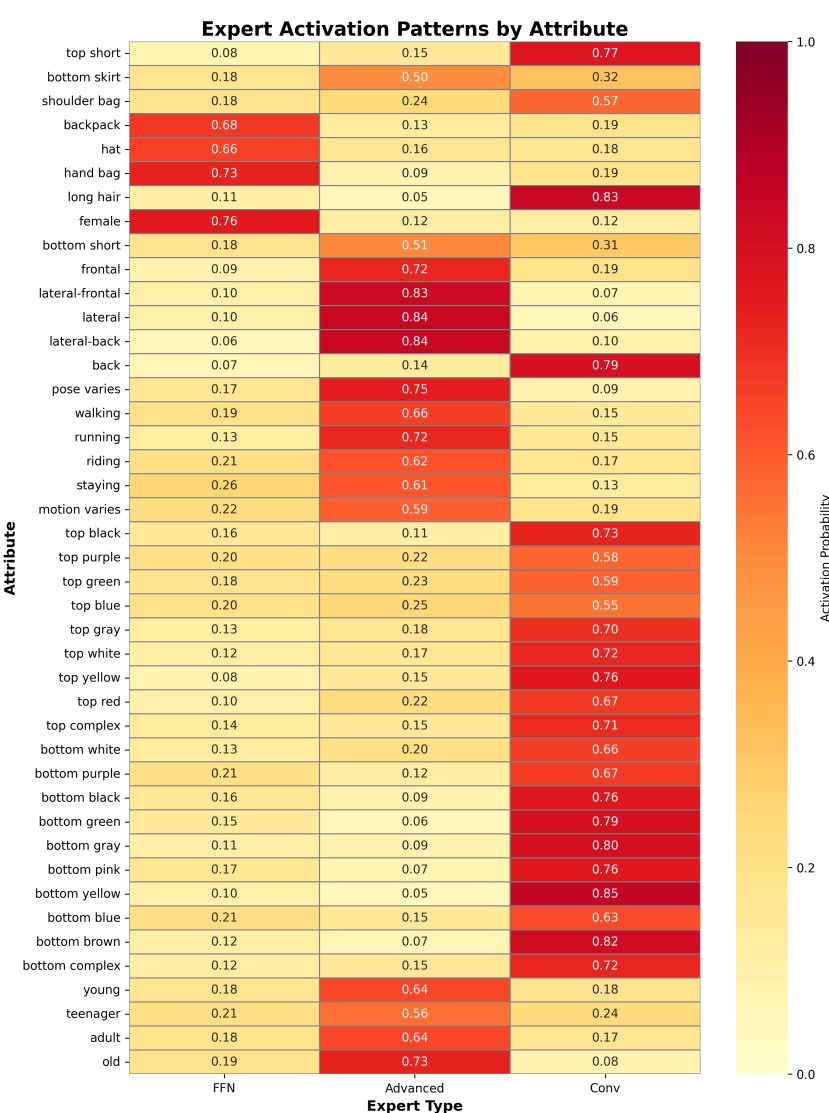

Figure 3: Expert activation patterns across attributes. The three expert types exhibit clear specialization: Conv experts strongly activate for localized appearance cues (e.g., colors, accessories), Advanced experts focus on pose and orientation attributes, and FFN experts contribute more weakly and broadly. These distinct patterns highlight that VLAAR encourages complementary, semantically aligned expert behaviors that improve attribute recognition.

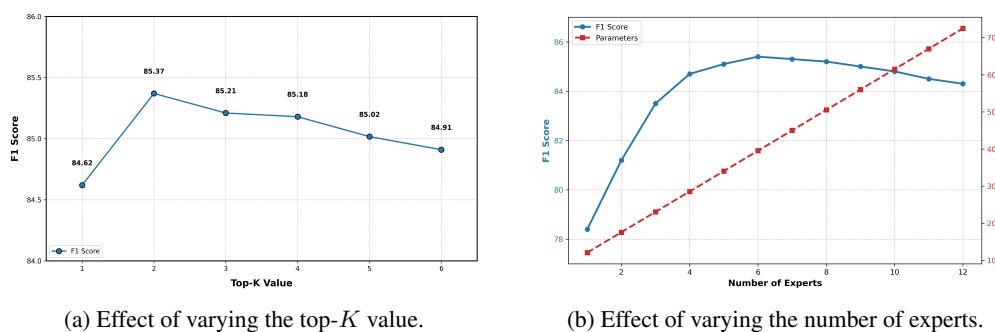

(a) Effect of varying the top-$K$ value.  (b) Effect of varying the number of experts.

Figure 4: Ablation studies on key hyperparameters. The best performance is achieved with $K = 2$ and 6 experts.

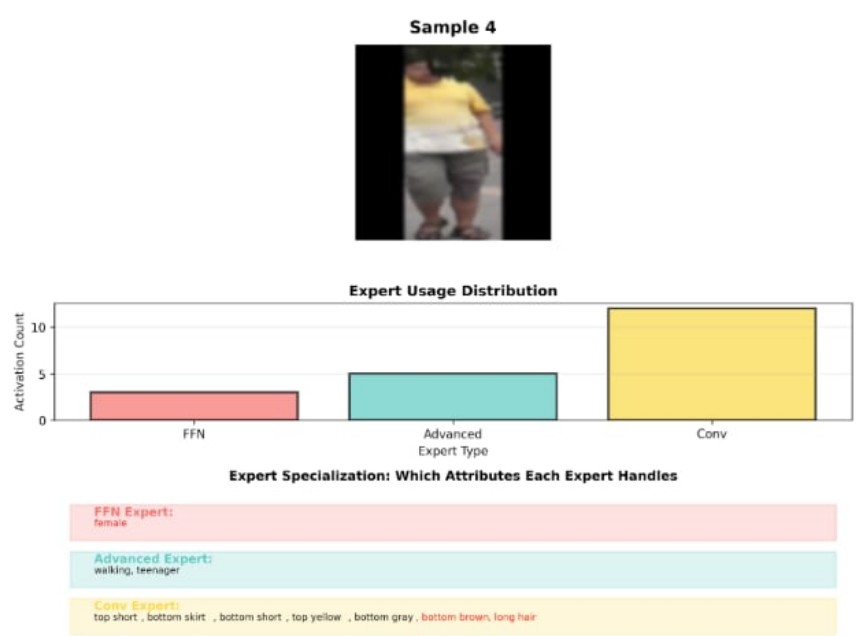

Figure 5: Failure cases of VLAAR on low-resolution, occluded surveillance images. The model struggles when visual features are heavily degraded, highlighting the need for feature enhancement or upscale-aware experts.

block, enabling the model to refine visual features and perform attribute recognition simultaneously. This would make the system more robust when handling inherently low-quality images.