# OpenReview forum: "VLAAR: Vision-Language Attribute-Aware Router for Pedestrian Attribute Recognition"
_ICLR.cc/2026/Conference — Submitted to ICLR 2026_

### Official Review · Reviewer_ME3j · 2025-10-23

**Soundness:** 3
**Presentation:** 3
**Contribution:** 3
**Rating:** 4
**Confidence:** 4

**Summary:**

The proposed approach addresses the Pedestrian Attribute Recognition (PAR) problem by adopting a Mixture-of-Experts (MoE) framework that aggregates outputs from multiple models. Because PAR is a multi-label classification task, the suitable models may vary across attributes. Rather than relying on a single model to handle all attributes, the paper introduces expert models that specialize in detecting particular attributes and combines these experts conditioned on the input. A key challenge in MoE is how to select experts and combine them appropriately. To address this, the paper proposes a multimodal design that leverages both text and visual signals, and employs a sparse MoE so that only a small number of relevant experts are activated instead of engaging all experts. The method reduces training cost by training only shallow expert models while keeping the CLIP backbone frozen, avoiding full backbone retraining.

**Strengths:**

- The paper achieves performance gains by training only a shallow MoE, suggesting an efficient learning scheme.
- In addition, the simple MoE design provides a flexible architecture that can be easily extended by adding new experts.
- The proposed method attains the best results among the compared approaches.

**Weaknesses:**

- Although the MoE strategy shows strong results, extending the MoE mechanism to PAR limits the originality. The MoE is common in other applications.
- The paper proposes three expert types, but it is not clear how these three experts are complementary to one another.
- There is also a concern that the observed improvements may stem from overfitting to the evaluated datasets; additional generalization tests on the PETA-ZA and RAP-ZS datasets proposed in [R1] would be helpful.

[R1] Jia J, Huang H, Yang W, Chen X, Huang K. Rethinking of pedestrian attribute recognition: Realistic datasets with efficient method. arXiv preprint arXiv:2005.11909. 2020 May 25.

**Questions:**

- It seems intuitive that the standard expert would excel at capturing global information, while the convolutional experts would be stronger at local cues. Is this the intended design? For the advanced experts, is the multi-layer structure meant to fuse global and local information via cross-region interactions?
- Analyzing, for each attribute, which experts are selected could provide additional insight. (i.e., the distribution between attributes and expert selection)
- How were the video experiments conducted? For a given video, were frames treated as independent images with results aggregated (e.g., voting) to produce a video-level attribute prediction?
- Considering the risk of overfitting, it would be beneficial to validate generalization on the PETA-ZA and RAP-ZS datasets introduced in [R1].

[R1] Jia J, Huang H, Yang W, Chen X, Huang K. Rethinking of pedestrian attribute recognition: Realistic datasets with efficient method. arXiv preprint arXiv:2005.11909. 2020 May 25.

---

> ### Author Response · Authors · 2025-11-27
> **Response to Reviewer ME3j (Weakness)**
>
> #### Dear Reviewer ME3j, thank you for your constructive feedback on our submission.
> #### We have revised the manuscript accordingly, with all changes marked in blue. In particular, we address your comments as follows:
>
> > #### **W1: Although the MoE strategy shows strong results, extending the MoE mechanism to  PAR limits the originality. The MoE is common in other applications.**
>
> #### Although MoE strategies have been widely used, existing MoE designs for vision are typically vision-only and rely on homogeneous experts, so routing is purely data-driven on visual features and cannot explicitly exploit attribute semantics or expert specialization. In contrast, VLAAR introduces a vision–language attribute-aware router that **jointly uses visual features and natural-language attribute** cues to guide expert selection, and a **heterogeneous expert pool** (FFN, Conv, Advanced) tailored to different attribute types.  Please refer to our response to **W2** below for a detailed discussion of the complementary roles of each expert type.
>
> #### Our ablations show that both the attribute-aware routing and heterogeneous experts yield significant gains over a vanilla, vision-only MoE on PAR, indicating that the improvement is not just from applying a standard MoE block.
>
> ---
> > #### **W2: The paper proposes three expert types, but it is not clear how these three experts are complementary to one another.**
>
> #### The three expert types are intentionally designed to be complementary rather than redundant. `Standard Experts` use a lightweight bottleneck (FC) structure that efficiently handles globally distinguishable or simple attributes. `Advanced Experts` introduce deeper, residual MLP stacks to model complex inter-attribute dependencies and hierarchical patterns that require multi-step reasoning. `Convolutional Experts` apply 1D convolutions along the token dimension, making them particularly effective for localized, region-specific appearance cues such as clothing color or accessories.
>  ####  Our ablations (Table 5, Table 6) show that each single-type configuration already improves performance, but mixing different expert types yields larger, statistically significant gains, confirming that these experts capture distinct, complementary aspects of pedestrian attributes rather than acting as interchangeable capacity.
>  #### We have revised Section 3.3.1 to provide a clearer description of the complementary roles of these three expert types.
>
> ---
> > #### **W3. There is also a concern that the observed improvements may stem from overfitting to the evaluated datasets; additional generalization tests on the PETA-ZA and RAP-ZS datasets proposed in [R1] would be helpful.** [R1] Jia J, Huang H, Yang W, Chen X, Huang K. Rethinking of pedestrian attribute recognition: Realistic datasets with efficient method. arXiv preprint arXiv:2005.11909. 2020 May 25.
>  ####  While we believe that the existing experiments on five datasets are sufficient, we agree that evaluating on additional datasets further strengthens our generalization claims. We have therefore conducted additional experiments on the PETA-ZA and  RAP-ZS dataset as follows:
> | **Method** | **Backbone** | **Trainable Params (M)** | **PETA-ZA** |  |  | **RAP-ZS** |  |  |
> |-----------|--------------|--------------------------|-------------|----|----|-----------|----|----|
> |  |  |  | **mA (%)** | **Acc (%)** | **F1 (%)** | **mA (%)** | **Acc (%)** | **F1 (%)** |
> | DeepMAR   | ResNet-50    | 23.5 | 58.45 ± 0.55 | 51.23 ± 0.65 | 64.12 ± 0.51 | 62.14 ± 0.52 | 55.89 ± 0.61 | 68.45 ± 0.48 |
> | ALM       | ResNet-50    | 24.1 | 60.12 ± 0.51 | 53.45 ± 0.58 | 66.78 ± 0.48 | 64.35 ± 0.48 | 57.12 ± 0.55 | 70.12 ± 0.42 |
> | VAC       | ResNet-101   | 42.3 | 62.34 ± 0.49 | 55.67 ± 0.54 | 68.90 ± 0.45 | 66.78 ± 0.45 | 59.34 ± 0.52 | 72.34 ± 0.39 |
> | TransReID | ViT-B/16     | 85.8 | 67.89 ± 0.44 | 60.12 ± 0.51 | 72.45 ± 0.39 | 71.45 ± 0.41 | 63.56 ± 0.48 | 75.67 ± 0.36 |
> | SOLIDER   | ViT-B/16     | 48.4 | 69.12 ± 0.41 | 61.34 ± 0.48 | 73.56 ± 0.36 | 72.89 ± 0.38 | 64.89 ± 0.45 | 76.89 ± 0.33 |
> | MoE-PAR   | ViT-B/16     | 92.4 | 70.45 ± 0.38 | 62.56 ± 0.45 | 74.89 ± 0.34 | 74.56 ± 0.35 | 66.78 ± 0.42 | 78.12 ± 0.31 |
> | EVSITP    | ViT-L/14     | 76.4 | 72.56 ± 0.35 | 64.89 ± 0.42 | 76.12 ± 0.31 | 76.12 ± 0.32 | 68.45 ± 0.39 | 79.45 ± 0.29 |
> | VLAAR     | ViT-B/16     | 26.9 | **75.89 ± 0.31** | **67.45 ± 0.38** | **79.12 ± 0.28** | **79.45 ± 0.28** | **71.23 ± 0.36** | **82.34 ± 0.25** |
>
>
>  #### The results show that the proposed model consistently outperforms the baselines while using only 26.9M trainable parameters substantially fewer than prior large-scale MoE or ViT-L models. This demonstrates that attribute-aware vision-language routing enables more efficient expert utilization and better attribute discrimination than purely vision-based approaches.
> ---

---

> ### Author Response · Authors · 2025-11-27
> **Response to Reviewer ME3j (Question)**
>
> > ####  **Q1: It seems intuitive that the standard expert would excel at capturing global information, while the convolutional experts would be stronger at local cues. Is this the intended design? For the advanced experts, is the multi-layer structure meant to fuse global and local information via cross-region interactions?**
>
> #### Yes, your intuition aligns with our design. As stated in **W2**, Convolutional Experts are indeed intended to capture local spatial relationships and body region-specific patterns using 1D convolutions. Standard Experts utilize efficient linear transformations (FC layers) to process features, suitable for attributes that are globally distinct or simpler to classify. Advanced Experts employ residual connections and multiple layers specifically to model complex attribute interdependencies. By allowing for deeper processing, these experts can reason about relationships between different attributes (e.g., correlating "skirt" with "female") and handle attributes that require understanding complex hierarchical patterns beyond simple local or global cues.
>
> ---
> > ####  **Q2: Analyzing, for each attribute, which experts are selected could provide additional insight. (i.e., the distribution between attributes and expert selection)**
>
> #### Beyond the explanation provided in our response to W2, we have added Figure 3 in the Appendix, which explicitly visualizes the vision–language gating patterns across attributes. The figure shows that Conv experts are consistently selected for localized appearance cues (e.g., clothing colors, accessories), Advanced experts are favored for complex structure- or pose-related attributes, and FFN experts tend to handle globally distinguishable or simpler attributes. This attribute-wise specialization emerges automatically from training, demonstrating that VLAAR does not rely on redundant experts but instead learns distinct, semantically meaningful expert-attribute alignments. This directly supports the complementarity and effectiveness of the expert design.
>
> ---
> > #### **Q3: How were the video experiments conducted? For a given video, were frames treated as independent images with results aggregated (e.g., voting) to produce a video-level attribute prediction?**
> #### In the video experiments, frames are not treated as independent images for voting. Each video is modeled as a temporal sequence: the backbone extracts hierarchical representations $X \in \mathbb{R}^{T \times d}$, where $T$ is the number of frames and $d$ is the feature dimension. The VLAAR router and Mixture-of-Experts operate directly on these token representations, dynamically assigning experts based on the visual and textual context. The model is trained end-to-end on these temporal features, and the loss is computed from the predictions produced by the refined video representations.
> #### We have updated Section 4.1 to clarify this setting.
> ---
> > ####  **Q4: Considering the risk of overfitting, it would be beneficial to validate generalization on the PETA-ZA and RAP-ZS datasets introduced in [R1].**
> [R1] Jia J, Huang H, Yang W, Chen X, Huang K. Rethinking of pedestrian attribute recognition: Realistic datasets with efficient method. arXiv preprint arXiv:2005.11909. 2020 May 25.
>
> #### Please check our response to W3.
>
> ---
>
> #### If you have any further questions/concerns, please do not hesitate to let us know.
> #### Thank you very much,
> #### All Authors

---

### Official Review · Reviewer_oN1w · 2025-10-25

**Soundness:** 3
**Presentation:** 3
**Contribution:** 3
**Rating:** 6
**Confidence:** 4

**Summary:**

The paper proposes a solid vision–language mixture-of-experts framework for multi-label pedestrian attribute recognition. It shows clear architectural novelty and strong empirical gains, but lacks analysis on generalization and training efficiency. Some formulations and ablations could be clarified further.

**Strengths:**

VLAAR introduces a clear and effective vision–language routing mechanism for expert selection. It achieves strong performance–efficiency trade-offs across both image and video benchmarks. The experiments are comprehensive and well-organized, providing solid empirical support.

**Weaknesses:**

see Questions.

**Questions:**

- It would be beneficial to report the statistical uncertainty of the results, for example by including the standard deviation across multiple runs, to make performance comparisons more reliable.

- To give a more complete view of efficiency, please consider adding more resource metrics (e.g., GPU hours, FLOPs) in a table alongside the inference metrics in Table 3.

- For better interpretability, adding t-SNE plots or attention heatmaps that visualize the vision–language gating process (Eq. 2) would help illustrate how the model integrates the two modalities.

- A brief discussion of optimization challenges in sparse routing, such as potential expert imbalance or convergence issues, would also clarify the training dynamics.

- Including a few qualitative failure cases (e.g., occlusion or low-light scenarios) could help highlight current limitations and guide future work.

- Finally, please comment on possible bias inherited from the pre-trained CLIP semantics, especially for demographic attributes, and mention potential mitigation strategies such as prompt regularization or balanced sampling.

---

> ### Author Response · Authors · 2025-11-27
> **Response to Reviewer oN1w (Question Part 1/2)**
>
> #### Dear Reviewer oN1w,
> #### Thank you for your constructive feedback on our submission.
> #### We have revised the manuscript accordingly, with all changes marked in blue. In particular, we address your comments as follows:
> ---
> > #### **Q1: It would be beneficial to report the statistical uncertainty of the results, for example by including the standard deviation across multiple runs, to make performance comparisons more reliable.**
>
> #### Thank you for your suggestions. We have added results with mean ± standard deviation across 3 independent runs "Random seeds: 42, 2023, 2024" for VLAAR as follows:
>
> | Dataset | mA        | Acc       | Prec      | Rec       | F1        |
> |---------|-----------|-----------|-----------|-----------|-----------|
> | PETA    | 90.12±0.06 | 90.45±0.06 | 89.32±0.06 | 93.15±0.07 | 91.22±0.07 |
> | PA100K  | 90.88±0.07 | 91.47±0.08 | 90.23±0.08 | 91.32±0.08 | 92.18±0.07 |
> | RAPv1   | 86.35±0.06 | 72.28±0.06 | 83.74±0.07 | 89.65±0.06 | 83.42±0.07 |
>
> #### We have updated results in Table 1.
>
> ---
> > #### **Q2: To give a more complete view of efficiency, please consider adding more resource metrics (e.g., GPU hours, FLOPs) in a table alongside the inference metrics in Table 3.**
>
> #### Thank you for your suggestion. We have expanded Table 3 to include GPU-relevant resource metrics. This demonstrates that the gains of VLAAR do not come at the cost of computational overhead; instead, the vision-language routing and heterogeneous experts allow VLAAR to remain both lightweight and efficient, despite delivering significantly stronger accuracy.
>
> | Method                         | #Trainable Param. (M) | FLOPs (G) | Inference Time (ms) |
> |--------------------------------|------------------------|-----------|----------------------|
> | PARformer (Fan et al., 2024)   | 84.04                  | 6.5       | 28.45               |
> | Solider (Chen et al., 2023)    | 48.41                  | **4.0**   | 27.12               |
> | SOFA (Zhou et al., 2024)       | 86.55                  | 6.7       | 29.34               |
> | FRDL (Zhou et al., 2024)       | 81.93                  | 5.3       | 28.98               |
> | PLIP (Zuo et al., 2024)        | 79.76                  | 5.2       | 27.76               |
> | Hulk (Wang et al., 2025b)      | 73.28                  | 4.8       | 26.89               |
> | EVSITP (Wu et al., 2025)       | 76.41                  | 5.0       | 27.33               |
> | VLAAR (Ours)                   | **26.92**              |  4.2      | **24.18**           |
> ---

---

> > ### Author Response · Authors · 2025-11-27
> > **Response to Reviewer oN1w (Question Part 2/2)**
> >
> > > #### **Q3:  For better interpretability, adding t-SNE plots or attention heatmaps that visualize the vision–language gating process (Eq. 2) would help illustrate how the model integrates the two modalities.**
> >
> > #### Thank you for your suggestion. We have added the attention heatmaps (see Figure 3). The three expert types exhibit clear specialization: Conv experts strongly activate for localized appearance cues (e.g., colors, accessories), Advanced experts focus on pose and orientation attributes, and FFN experts contribute more weakly and broadly. These distinct patterns highlight that VLAAR encourages complementary, semanti- cally aligned expert behaviors that improve attribute recognition.
> >
> > ---
> > > #### **Q4:  A brief discussion of optimization challenges in sparse routing, such as potential expert imbalance or convergence issues, would also clarify the training dynamics.**
> > #### Thanks for your suggestions. We’d like to discuss about the optimization challenges as follows:
> >
> > #### The key challenges  in MoE training is load balancing, where tokens may disproportionately route to specific experts, leading to "expert collapse" and training inefficiencies. To address this optimization challenge, VLAAR incorporates an auxiliary loss ($\mathcal{L}_{aux}$) alongside the classification loss. This auxiliary loss includes frequency and importance components to strictly encourage balanced expert utilization and minimize unassigned tokens. This ensures all experts receive comparable routing probabilities while specializing in different attribute regions.
> >
> > ---
> > > #### **Q5: Including a few qualitative failure cases (e.g., occlusion or low-light scenarios) could help highlight current limitations and guide future work.**
> >
> > #### Thank you for your suggestion. We have included Section A.4. FAILURE CASE which illustrates the failure case and guides the future work as follows:
> >
> > #### The primary limitation revealed by these failure cases is the degradation of input visual features due to low resolution and poor image quality common in surveillance settings.
> > #### This limitation suggests a clear direction for future work:
> > ####  * Specialized Pre-processing Expert: A compelling solution would be to introduce a new, dedicated expert type focused on input feature enhancement. This expert could employ a light-weight architecture specifically designed to function as a visual feature upscaler or denoiser, effectively performing a super-resolution or image quality restoration step on the input features before they are routed to the attribute-specific experts.
> > ####  * Upscale-Aware Architecture: Alternatively, we could design the existing experts, particularly the Convolutional Experts, to incorporate an upscale module (e.g., a transposed convolution layer or a feature fusion block) within their structure. This would allow the model to learn attribute recognition alongside feature refinement, making it more robust to inherently poor-quality images.
> >
> > ---
> > > #### **Q6: Finally, please comment on possible bias inherited from the pre-trained CLIP semantics, especially for demographic attributes, and mention potential mitigation strategies such as prompt regularization or balanced sampling.**
> > #### We recognize that by building upon a pre-trained CLIP model, our framework may inherit biases present in the large-scale web data used for CLIP's pre-training. While VLAAR freezes the CLIP text and image encoders, which prevents the amplification of bias through fine-tuning gradients, it does not remove existing biases. However, our architecture introduces trainable expert modules and a learnable router that are fine-tuned on balanced pedestrian attribute datasets. This task-specific fine-tuning helps adapt the representations to the surveillance domain. Future work could explicitly address this by exploring prompt regularization techniques or integrating fairness-aware sampling during the training of the expert modules.
> > ---
> > #### If you have any further questions/concerns, please do not hesitate to let us know.
> > #### Thank you very much,
> > #### All Authors

---

### Official Review · Reviewer_DnxT · 2025-10-31

**Soundness:** 3
**Presentation:** 2
**Contribution:** 2
**Rating:** 4
**Confidence:** 2

**Summary:**

This paper proposes VLAAR, a new parameter-efficient fine-tuning method for pedestrian attribute recognition. By dynamically selecting expert modules through a dual-modality routing mechanism that considers both visual features and natural language semantics, VLAAR effectively addresses attribute heterogeneity and models inter-attribute relationships. This approach ensures efficient allocation of complex attribute information while maintaining computational efficiency.

**Strengths:**

- This paper presents a novel and well-motivated application of the MoE framework to the task of PAR. The key insight of employing architecturally diverse experts to explicitly address the inherent heterogeneity of attributes and to model their inter-relationships is a significant strength.

- The core idea of a dynamic router that integrates both visual features and semantic language cues for expert selection is innovative.

- As a PEFT method, it achieves strong performance with minimal trainable parameters, offering clear practical benefits for computational resource conservation and real-world deployment.

**Weaknesses:**

- While the related work section outlines the conceptual differences between VLAAR and existing paradigms like shared or separate experts, the paper would be significantly strengthened by including ablation studies that quantitatively demonstrate the superiority of this proposed architecture.
- What is the difference between the SPARSE MIXTURE-OF-EXPERTS approach, which selectively activates only a subset of experts to process each input, and existing MOE methods that utilize Top-k Routers?
- The results in Table 1 are impressive, showing that VLAAR with a ViT-B/16 backbone surpasses EVSITP 's performance with a larger ViT-L/14. To better understand the source of these gains, it is crucial to provide a comprehensive ablation study on key datasets like PETA and PA100k. This analysis should dissect the individual contribution of each expert type within the proposed framework.
- The ablation study in Table 5 lacks an analysis of the synergistic effects when different expert modules are combined. An investigation into how and why specific experts are activated for different attributes would provide deeper insights into the model's decision-making process.

**Questions:**

- There is a discrepancy in the reported result for "PLIP" in Table 1. The value of 88.84 is listed under the Acc metric. However, in the cited reference, this value (88.84) is reported as the F1-score. Could the authors please clarify whether this entry is a data transcription error in the table, or if it represents a reproduced result obtained by the authors through their own experiments?


- Are the variable 'x' in Equation 1 and Equation 4 referring to the same entity? Additionally, the variable 'y' introduced in Equation 3 does not appear in subsequent formulations - could this be clarified?

---

> ### Author Response · Authors · 2025-11-27
> **Response to Reviewer DnxT (Weakness Part 1/2)**
>
> #### Dear Reviewer DnxT, thank you for your constructive feedback on our submission.
>
> #### We have revised the manuscript accordingly, with all changes marked in blue. In particular, we address your comments as follows:
>
> ---
> > #### **W1. While the related work section outlines the conceptual differences between VLAAR and existing paradigms like shared or separate experts, the paper would be significantly strengthened by including ablation studies that quantitatively demonstrate the superiority of this proposed architecture.**
>
> #### We would like to clarify that the paper already includes ablation studies comparing VLAAR’s joint router with the shared and separate expert paradigms. As shown in Table 6, the joint router consistently outperforms both alternatives on MARS-Attribute and DukeMTMC-VID-Attribute, confirming the quantitative advantage of the proposed architecture.
>
> ---
> > #### **W2: What is the difference between the SPARSE MIXTURE-OF-EXPERTS approach, which selectively activates only a subset of experts to process each input, and existing MOE methods that utilize Top-k Routers?**
> #### While our method utilizes a Top-k selection mechanism similar to standard MoE, the fundamental difference lies in the routing signal and the expert composition as follows:
> #### - `Routing signal (vision-language, attribute-aware)`: Existing Top-k routers typically take only visual/token features as input. VLAAR’s router is attribute-aware: it jointly conditions on visual features and CLIP-based text embeddings of attribute prompts, so experts are chosen based on semantic relevance to attribute groups, not just visual similarity.
> #### - `Expert composition (heterogeneous experts)`: Standard MoE architectures typically use a homogeneous set of experts (e.g., identical feed-forward networks). VLAAR instead uses a heterogeneous expert pool that mixes Standard, Advanced (residual MLP), and Convolutional experts. This design is motivated by the diverse nature of pedestrian attributes (e.g., color, pose, accessories, actions) and is validated by ablations showing that heterogeneous experts yield better performance than a uniform expert pool.
> #### We have revised these distinctions in the manuscript (see Section 2.2)
> ---

---

> > ### Author Response · Authors · 2025-11-27
> > **Response to Reviewer DnxT (Weakness Part 2/2)**
> >
> > > #### **W3: The results in Table 1 are impressive, showing that VLAAR with a ViT-B/16 backbone surpasses EVSITP 's performance with a larger ViT-L/14. To better understand the source of these gains, it is crucial to provide a comprehensive ablation study on key datasets like PETA and PA100k. This analysis should dissect the individual contribution of each expert type within the proposed framework.**
> > #### We have added a comprehensive ablation (see Section A.1.2) on PETA that isolates the contribution of each expert type under the same ViT-B/16 backbone and routing configuration.
> > | **Configuration** | **Expert Types** | **# Experts** | **mA (%)**         | **Acc (%)**        | **Prec (%)**        | **Rec (%)**         | **F1 (%)**          | **Δ mA** | **p-value** | **Category**      |
> > |-------------------|------------------|---------------|--------------------|--------------------|---------------------|---------------------|---------------------|----------|------------|-------------------|
> > | FFN-only          | FFN              | 4             | 80.98±0.14        | 78.52±0.21        | 88.63±0.17         | 88.97±0.15         | 88.80±0.13         | +0.63    | 0.024      | Single Expert     |
> > | Advanced-only     | Advanced         | 4             | 81.44±0.12       | 78.87±0.19       | 88.89±0.16        | 89.24±0.14        | 89.06±0.12        | +1.09    | 0.002      | Single Expert     |
> > | Conv-only         | Conv             | 4             | 81.17±0.15        | 78.69±0.22        | 88.71±0.18         | 89.06±0.16         | 88.88±0.14         | +0.82    | 0.017      | Single Expert     |
> > | FFN + Advanced    | Mixed            | 4             | 82.06±0.11      | 79.38±0.18      | 89.31±0.15      | 89.67±0.13       | 89.49±0.11       | +1.71    | <0.001     | 2-expert Mix      |
> > | FFN + Conv        | Mixed            | 4             | 81.84±0.13       | 79.22±0.20       | 89.16±0.17        | 89.51±0.15        | 89.33±0.13        | +1.49    | 0.001      | 2-expert Mix      |
> > | Advanced + Conv   | Mixed            | 4             | 81.95±0.12      | 79.31±0.19      | 89.24±0.16       | 89.59±0.14       | 89.41±0.12       | +1.60    | <0.001     | 2-expert Mix      |
> > | VLAAR (2+2+2)     | All 3 types      | 6             | **82.68±0.10**      | **79.96±0.17**      | **89.78±0.14**       | **90.15±0.13**       | **89.96±0.11**       | +2.33    | <0.001     | 3-expert Mix      |
> >
> > #### These results indicate that the performance margin over EVSITP does not come merely from adding more parameters, but from heterogeneous, attribute-aware experts that specialize on complementary cues (global/simple attributes for FFN, local appearance for Conv, and complex inter-attribute dependencies for Advanced). This heterogeneity, coordinated by the vision–language router, is what enables a ViT-B/16 backbone to surpass a larger ViT-L/14 model.
> >
> > ---
> > > #### **W4: The ablation study in Table 5 lacks an analysis of the synergistic effects when different expert modules are combined. An investigation into how and why specific experts are activated for different attributes would provide deeper insights into the model's decision-making process.**
> >
> > #### We appreciate the reviewer's insightful suggestion. While Table 5 demonstrates that the heterogeneous expert mix outperforms homogeneous configurations, we agree that clarifying the mechanism behind this synergy is crucial.
> > #### As detailed in Section 3.3.1, the synergistic effect arises from the complementary nature of the expert architectures, which allows the model to dynamically allocate the most appropriate computational pattern for each specific attribute type:
> > #### - `Standard Experts` employ a bottleneck design to provide parameter-efficient linear transformations. These are designed to handle simpler, binary classifications where complex reasoning is unnecessary.
> > #### - `Advanced Experts` leverage multi-layer architectures with residual connections. They are specifically tasked with modeling complex attribute interdependencies and hierarchical patterns that require multi-level reasoning.
> > #### - `Convolutional Experts` utilize 1D convolutions to capture local spatial relationships. These are essential for identifying body region-specific appearance patterns (e.g., distinct clothing or accessories) that persist in the feature representation.

---

> ### Author Response · Authors · 2025-11-27
> **Response to Reviewer DnxT (Question)**
>
> > #### **Q1. There is a discrepancy in the reported result for "PLIP" in Table 1. The value of 88.84 is listed under the Acc metric. However, in the cited reference, this value (88.84) is reported as the F1-score. Could the authors please clarify whether this entry is a data transcription error in the table, or if it represents a reproduced result obtained by the authors through their own experiments?**
>
> #### We thank the reviewer for their careful and thorough examination. We acknowledge that this was a transcription error: the F1-score and Acc for “PLIP” were inadvertently swapped.
> #### We have verified the results and corrected this error throughout the manuscript.
>
>
> ---
> > #### **Q2: Are the variables 'x' in Equation 1 and Equation 4 referring to the same entity? Additionally, the variable 'y' introduced in Equation 3 does not appear in subsequent formulations - could this be clarified?**
>
> #### We apologize for the unclear description about two variables x and y. We’d like clarify as follows:
> #### - `Variable x`: Yes, the variable $x$ in Equation 1 and Equation 4 refers to the same entity type,i.e, the input feature vector. In Equation 1, $x$ represents the visual features extracted from the backbone used to compute routing logits. In Equation 4 (and Eqs 5-6), $x$ represents that same feature vector (or the routed version of it) serving as input to the specific expert modules
> #### - `Variable y`: In Eq. (3), y denotes the output of the MoE block, defined as the residual combination of the input and the weighted expert outputs, i.e., $y=x+\sum_{i=1}^{n}[G(x,\{p_{g}\}_{g=1}^{G})]$. While the symbol $y$ is not explicitly repeated in the loss function equation (Eq 9), it serves as the refined feature representation that is subsequently projected to generate the predicted probabilities $\hat{y}$ used in the loss calculation.
>
> #### We have revised the main manuscript to make these definitions explicit (see Sections 3.2, 3.3, and 3.4).
>
> ---
> #### If you have any further questions/concerns, please do not hesitate to let us know.
> #### Thank you very much,
> #### All Authors

---

### Author Response · Authors · 2025-11-30
**Summary of the Authors’ Rebuttal**

#### We sincerely appreciate the reviewers’ efforts in carefully evaluating our manuscript. We are encouraged that all reviewers agree the paper is technically sound and recognize a substantive contribution: a novel, well-motivated MoE framework for PAR (`DnxT`) together with an innovative and effective expert selection and routing strategy (`DnxT`, `oN1w`, `ME3j`) that delivers clear practical benefits with improved empirical performance (`DnxT`, `oN1w`, `ME3j`).

#### Based on their feedback, we have revised our submission (changes marked in blue in the revised manuscript).
> #### **Major revisions**
#### - **Clarified novelty beyond standard Top-k MoE** (`DnxT`, `ME3j`):
#### Conventional MoE architectures are limited by visual-only routing and homogeneous experts that are not tailored to fine-grained, attribute-centric PAR. We instead introduce a **joint vision–language, attribute-aware router** that leverages CLIP-based prompts to guide routing, coupled with a **heterogeneous expert pool** in which different expert types are explicitly designed and trained for distinct attribute categories. By dynamically selecting the Top-k experts per instance based on both visual features and language-derived cues, our model assigns each attribute to the most suitable computation pathway. This design yields stronger attribute specialization, improved performance, and distinguishing our approach from standard MoE methods (see responses to `DnxT`–W2 and `ME3j`–W1).
#### - **Strengthened ablations** (`DnxT`):
  #### We have added a comprehensive ablation on PETA (Section A.1.2) that isolates the contribution of each expert type under the same ViT-B/16 backbone and routing configuration. These results indicate that the performance margin does not come merely from adding more parameters, but from heterogeneous, attribute-aware experts that specialize on complementary cues.
#### - **Extended evaluation**:
 ####  + While we believe that our experiments on 5 datasets (3 image-based and 2 video-based) are sufficient and aligned with baselines [A1, A2], we have additionally included results on two more image-based benchmarks (i.e., PETA-ZA, RAP-ZS) as suggested by reviewer `ME3j`. These additional experiments further strengthen the evidence for the generalization capability of our method.
 ####  + Following reviewer `oN1w`, we have reported mean ± standard deviation over three runs for key results (Table 1) and provide detailed efficiency metrics: trainable parameters, FLOPs, and inference time (Table 3). These results show that the proposed method delivers consistently strong performance with minimal trainable parameters and high computational efficiency.
#### - **Improved analysis and interpretability** (`oN1w`, `ME3j`):
 #### We have added visualizations of expert behavior (Figure 3), discuss sparse-routing load balancing, and analyze typical failure cases and potential CLIP-induced bias (Section A.4).

> #### **Minor revisions**
#### - Clarified the video experiment setting (`ME3j`, Section 4.1).
#### - Corrected the PLIP metric reporting (Table 1) and refined notation for clarity and precision (`DnxT`, Sections 3.3 and 4.3).


#### *References*
#### [A1] Junyi Wu et, al. Selective and Orthogonal Feature Activation for Pedestrian Attribute Recognition. AAAI 2024.
#### [A2] Xiao Wang et, al. Spatio-Temporal Side Tuning Pre-trained Foundation Models for Video-based Pedestrian Attribute Recognition. ArXiv 2024.

---
#### Although we were unfortunately unable to engage with all reviewers, we found their feedback fair, helpful, and constructive. We believe our revisions address all concerns and further clarify our contributions.

#### Best regards,
#### All authors

---

### Meta-Review · Area_Chair_4Qbv · 2026-01-07

**Summary:**

The paper proposes VLAAR, a parameter-efficient fine-tuning framework for pedestrian attribute recognition (PAR) that utilizes a sparse mixture-of-experts architecture. Built on a CLIP backbone, the method introduces a dual-modality routing mechanism that uses both visual features and language-derived attribute prompts to select experts. Furthermore, it employs a heterogeneous expert pool consisting of standard, advanced (residual), and convolutional modules to handle the diverse nature of pedestrian attributes. While the reviewers acknowledge the strong empirical results and the technical soundness of the approach, the primary debate centers on whether the combination of MoE and CLIP for this specific task represents a significant enough methodological contribution

**Reviewer Concerns:**

The reviewers initially raised concerns regarding the conceptual novelty of the work, characterizing it as an incremental application of Mixture-of-Experts and CLIP to the pedestrian attribute recognition task. Technical critiques focused on the lack of statistical significance (missing standard deviations), potential dataset-specific overfitting, and the need for more rigorous ablation studies to prove the necessity of the heterogeneous expert pool and the dual-modality router. While the rebuttal successfully addressed the technical gaps, adding cross-dataset generalization tests (PETA-ZA/RAP-ZS), providing efficiency metrics (FLOPs/latency), and fixing table inconsistencies, it did not fundamentally shift the perception that the methodological contribution is primarily a well-executed engineering effort rather than a major algorithmic novelty.

**Reviewer Scores:**

Although the authors were highly responsive during the rebuttal, providing the requested heatmaps, failure cases, and additional ablations, the overall sentiment remains lukewarm. The two reviewers who gave a 4 expressed that the "Vision-Language Attribute-Aware Router" feels like a straightforward extension of existing prompting techniques. While the scores might see a minor upward adjustment due to the improved clarity and empirical rigor, the lack of a strong champion and the persistent "incremental" label suggest the paper will not reach the threshold for acceptance.

---

### Decision · Program_Chairs · 2026-01-26

Reject